# Guided Similarity Separation for Image Retrieval

**Chundi Liu**
Layer6 AI
chundi@layer6.ai

**Guangwei Yu**
Layer6 AI
guang@layer6.ai

**Cheng Chang**
Layer6 AI
jason@layer6.ai

**Himanshu Rai**
Layer6 AI
himanshu@layer6.ai

**Junwei Ma**
Layer6 AI
jeremy@layer6.ai

**Satya Krishna Gorti**
Layer6 AI
satya@layer6.ai

**Maksims Volkovs**
Layer6 AI
maks@layer6.ai

## Abstract

Despite recent progress in computer vision, image retrieval remains a challenging open problem. Numerous variations such as view angle, lighting and occlusion make it difficult to design models that are both robust and efficient. Many leading methods traverse the nearest neighbor graph to exploit higher order neighbor information and uncover the highly complex underlying manifold. In this work we propose a different approach where we leverage graph convolutional networks to directly encode neighbor information into image descriptors. We further leverage ideas from clustering and manifold learning, and introduce an unsupervised loss based on pairwise separation of image similarities. Empirically, we demonstrate that our model is able to successfully learn a new descriptor space that significantly improves retrieval accuracy, while still allowing efficient inner product inference. Experiments on five public benchmarks show highly competitive performance with up to 24% relative improvement in mAP over leading baselines. Full code for this work is available here: `https://github.com/layer6ai-labs/GSS`.

## 1 Introduction

Image retrieval is a fundamental problem in computer vision with a wide range of applications including image search [43, 12], medical image analysis [21], 3D scene reconstruction [14], e-commerce [16, 24] and surveillance [41, 35]. To cope with the tremendous volume of visual data, most image retrieval systems address the problem in two stages. First, images are mapped to descriptors that support efficient inner product retrieval. Recent advances in convolutional neural networks ushered significant progress in descriptor models based on deep learning [13, 33], largely replacing traditional local feature descriptors [37, 25, 29] due to better performance and efficiency. Following descriptor retrieval, second stage refines the retrieved set by considering manifold structure that is known to be important for visual perception [34, 39].

Robust image retrieval remains a challenging problem. Variations in view angle, lighting and occlusion make it challenging to design retrieval models that are robust to these artifacts. Many currently leading approaches borrow ideas from clustering where similar challenges exist. They aim to satisfy local consistency where images with nearby descriptors are relevant, and global consistency where images on the same descriptor manifold are also relevant [47]. Popular methods in this category include query expansion (QE) [8] and its variants [33], that combine descriptors form neighboring images pushing them closer together. Another popular direction is similarity propagation/diffusion [48, 9, 19] that apply random walk on the nearest neighbor graph. Similarity propagation can explore higher-order neighbors than QE, and better uncover the underlying image manifold. While effective, both QE and similarity propagation rely on hyper-parameters that need to be tuned by hand, and typically no learning is done. This limits the representational power of these models as they rely heavily on base descriptors.

Inspired by these directions, we propose a new retrieval model that is trained end-to-end to improve both local and global consistency. To promote local consistency we use a graph convolutional network [23] to encode neighbor information into image descriptors. This has a similar effect to QE where nearby descriptors share information. However, unlike QE that always operates in the same descriptor space, our model learns a *new* representation that utilizes higher order neighbor information to improve retrieval. We additionally introduce a novel loss function to optimize the proposed model in a fully unsupervised fashion. The proposed loss encourages pairwise clustering of similarity scores, and stems from ideas in manifold learning where analogous pairwise objectives were shown to successfully uncover robust low-dimensional manifolds [15, 26, 44]. Finally, we show that our model can successfully learn refined neighbor information from spatial verification, and introduce an approximate inference procedure to leverage this information without applying spatial verification at query time. Experiments on five public benchmarks show highly competitive performance with up to 24% relative improvement in mAP over leading baselines.

## 2  Related Work

Popular research direction in image retrieval that shares motivation with our work is similarity propagation [46, 31, 9, 19, 3]. As the name suggests, similarity propagation applies random walk to propagate similarities on a weighted nearest-neighbor graph generated through descriptor retrieval. Similarities are propagated repeatedly until a global stable state is achieved. Spatial verification [2] is often applied in conjunction with similarity propagation to refine the neighbor graph and reduce false positives. Our approach differs from similarity propagation in that instead of traversing the nearest-neighbor graph we directly encode it into image descriptors, learning a new descriptor space in the process.

**Clustering**  Similarity propagation is based on the idea of local and global consistency [47] which stems from clustering and related fields [5, 44]. The commonly used *cluster assumption* states that decision boundary should lie in regions of low density, and equivalently, related points should be connected by a path passing through high density region [5]. Based on this assumption, many clustering approaches optimize pairwise similarity between examples demonstrating that it indirectly achieves the desired separation effect [39, 10, 6]. This forms the basis of our loss function and learning scheme – we adapt the pairwise clustering paradigm to unsupervised image retrieval where there are no fixed clusters or cluster centroids.

Deep Embedded Clustering (DEC) [44] is the most similar clustering approach to our work. Authors of DEC propose to start with a reasonably good model, then alternate between generating cluster assignments and optimizing the model using high confidence assignments. They demonstrate that improvement is made at each iteration by learning from high confidence assignments which in turn improves low confidence ones. Our approach follows a similar framework where we alternate between updating descriptors and maximizing (minimizing) already confidently high (low) pairwise similarity scores. The main difference is that our loss is designed for retrieval, and operates purely on image pairs without any explicit cluster information.

**Manifold Learning**  Manifold learning aims to infer low dimensional representations for high dimensional input that capture the manifold structure. This area is highly relevant to image retrieval as global descriptors aim to achieve a similar goal. Here, popular methods include IsoMap [39] LLE [34], t-SNE [26] and LINE [38]. In image retrieval, a manifold learning methods that is most similar to our work is the recently proposed IME [45]. Analogous to our approach, IME learns a new descriptor space by reconstructing pairwise distances between images in the nearest neighbor graph. The major difference, however, is that IME aims to preserve the structure of the graph generated by the base descriptors. Given that base descriptors (and the resulting neighbor graph) can have many inaccuracies, we instead focus on *improving* pairwise descriptor distances between images by learning from confident predictions that are likely to be correct.

## 3  Approach

We follow the standard image retrieval set-up used in literature [9, 19]. Given a database of $n$ images $\mathcal{X} = \{x_1, ..., x_n\}$ and a query image $x_q$, the goal is to retrieve all relevant images from $\mathcal{X}$ for $x_q$. Analogous to previous work we assume that global image descriptors have been extracted, and each image is represented by a vector $x \in \mathbb{R}^d$ [19]. We define a $k$-nearest neighbor ($k$-

NN) graph $G_k = (\mathcal{X}, A_k)$ with nodes $\mathcal{X}$ and edges described by the symmetric adjacency matrix $A_k = (a_{ij}) \in \mathbb{R}^{n \times n}$:

$$a_{ij} = \begin{cases} x_i^\top x_j & \text{if } x_j \in \mathcal{N}_k(x_i) \vee x_i \in \mathcal{N}_k(x_j) \\ 0 & \text{otherwise,} \end{cases} \tag{1}$$

where $\mathcal{N}_k(x)$ is the set of $k$ nearest neighbors (including itself) of $x$ in the descriptor space. The adjacency matrix is highly sparse with no more than $2kn$ non-zero values. Currently leading image retrieval approaches use various versions of the $k$-NN graph as transition matrix in a random walk process also known as similarity propagation/diffusion [9, 19, 18]. Random walk enables effective traversal of the underlying manifold significantly improving retrieval quality over base descriptors. However, most similarity propagation models depend on hyper-parameters that need to be tuned by hand, and typically no learning is done. In this work we propose a different approach where we encode neighbor information directly into image descriptors, and then train the model to learn a new descriptor space with desired properties.

The main idea behind our approach stems from clustering which also forms the basis of many similarity propagation methods [47, 48]. Specifically, we aim to satisfy both local consistency where images with nearby descriptors are relevant, and global consistency where images on the same descriptor manifold are also relevant. To achieve this we propose a new architecture where graph convolutional network (GCN) [23] is used to encode information from the $k$-NN graph into image descriptors to achieve local consistency. We then introduce a novel loss function that encourages GCN to learn descriptors that improve global consistency. Our approach, referred to as **G**uided **S**imilarity **S**eparation (GSS), is fully unsupervised and supports efficient inner product retrieval.

### 3.1 Model Architecture

In similarity propagation, local consistency is achieved by defining a transition matrix based on the $k$-NN graph. Here, we take a different approach and use a GCN [23] architecture to achieve a similar effect. The main operation in each GCN layer is the multiplication of the (normalized) adjacency matrix $A_k$ with image descriptors $\mathcal{X}$. This has the effect of applying weighted average to neighbor descriptors. Analogous to query expansion [8], averaged descriptors move closer together improving local consistency. However, unlike similarity propagation that uses a static transition matrix, parameterized GCN iteratively learns an entirely new descriptor space. This significantly increases the representational power of the model, while maintaining efficient retrieval and storage enabled by global descriptors. To apply GCN in this setting, we first normalize the adjacency matrix using a similar procedure to [23]:

$$\tilde{a}_{ij} = \left( \sum_{m=1}^{n} a_{im} \right)^{-\frac{1}{2}} \left( \sum_{m=1}^{n} a_{jm} \right)^{-\frac{1}{2}} a_{ij} \tag{2}$$

Normalisation reduces bias towards "popular" images that appear in many $k$-NN neighborhoods, and improves optimization stability. Using the normalized matrix, a multi-layer GCN is then defined as:

$$h_i^{(l+1)} = \sigma \left( w^{(l)} \sum_j \tilde{a}_{ij} h_j^{(l)} + b^{(l)} \right) \tag{3}$$

where $h_i^{(l)}$ is the output of the $l$'th layer for image $x_i$ with $h_i^{(0)} = x_i$, and $\sigma$ is the non-linear activation function; $w^{(l)}$ and $b^{(l)}$ are weight and bias parameters to be learned. Note that we deviate from the standard GCN definition of [23] by introducing a bias term which we empirically found to be beneficial. Successive GCN layers capture increasingly higher order neighbor information by repeatedly combining descriptors for each image with its nearest neighbors. The output of the last layer $(L)$ is then normalized to produce new descriptors:

$$\tilde{x}_i = \frac{h_i^{(L)}}{\|h_i^{(L)}\|_2} \tag{4}$$

Once the network is trained, retrieval is done via inner product in the new descriptor space $\tilde{x}$.

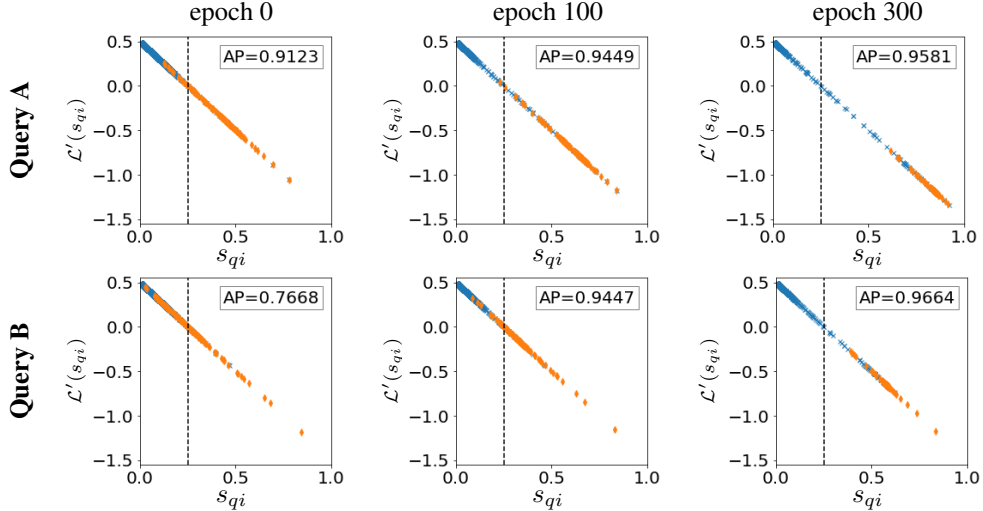

Figure 1: Top and bottom rows show two different $\mathcal{R}$Oxford queries A and B at various stages of training from start (epoch 0) to epoch 300. Each figure shows GSS gradient $\mathcal{L}'(s_{qi}) = -\alpha(s_{qi} - \beta)$ against the similarity scores $s_{qi}$ between query $x_q$ and database images $x_i \in \mathcal{X}$. Here, $\alpha = 2$ and $\beta = 0.25$ shown with a dashed vertical line. Scores for relevant to the query images are colored in orange, and average precision (AP) retrieval score is shown at each training stage. Note that this model is trained in a fully unsupervised fashion and does not see the relevance labels.

## 3.2 Guided Similarity Separation

To improve global consistency we propose a new unsupervised loss based on pairwise alignment between descriptors. Previous work on clustering/manifold learning demonstrated that complex low-dimensional manifolds can be successfully learned by optimizing pairwise distances between examples [15, 26, 44]. Retrieval bears many similarities to clustering – if all relevant images are "closer" in descriptor inner product than non-relevant images perfect retrieval is achieved. Consequently, we hypothesize that by optimizing pairwise distances between descriptors we can achieve an analogous effect, and learn a global low-dimensional descriptor manifold that improves retrieval.

This forms the basis of our loss function. We assume that the base descriptors are reasonably good, so that at the start of training similarity scores $s_{ij} = \tilde{x}_i^\top \tilde{x}_j$ are generally higher when images are relevant (we empirically demonstrate this to be true). The main idea behind guided similarity separation is to increase $s_{ij}$ if it is above a given threshold and lower it otherwise. This has a clustering effect where images with higher similarity scores move closer together, and those with lower scores get pushed further apart. In gradient-based learning one way to achieve this effect is through a loss function that has the following derivative:

$$\frac{\partial \mathcal{L}(s_{ij})}{\partial s_{ij}} = -\alpha(s_{ij} - \beta) \tag{5}$$

where $\beta \in (0, 1)$ is a similarity threshold and $\alpha > 0$ controls the slope. Solving the above differential equation leads to our GSS loss:

$$\mathcal{L}(s_{ij}) = -\frac{\alpha}{2}(s_{ij} - \beta)^2 \tag{6}$$

We then restrict the range of similarity scores to $[0, 1]$ by re-scaling $s_{ij}$ with $\max(0, s_{ij})$, and set the gradient to be zero at the boundaries i.e. $\frac{\partial \mathcal{L}(s)}{\partial s}\big|_0 = 0$ and $\frac{\partial \mathcal{L}(s)}{\partial s}\big|_1 = 0$. While this results in two discontinuities, it is analogous to the discontinuity in the commonly used ReLU function and has little effect on optimization.

The GSS loss has a 3-fold effect. First, similarity scores above $\beta$ get increased bringing the corresponding descriptors closer together. Second, scores below $\beta$ get lowered pushing descriptors further apart. Third, scores near $\beta$ have little gradient and remain largely unchanged. Moreover, the magnitude of the gradient increases linearly with the distance from $\beta$, and $\alpha$ controls the rate of increase. Jointly this has the effect where confident predictions (in either direction) become more confident, while unconfident predictions remain largely unchanged.

The proposed loss further emphasizes already confident predictions so initial similarity scores need to be reasonably good. To ensure that at the start of learning, forward passes through randomly initialized GCN don't adversely affect the scores, we carefully initialize parameters in each GCN layer. Specifically, we set all biases to 0 and initialize weights $w^{(l)}$ to have unit diagonal with off-diagonal elements sampled from $\mathcal{N}(0, \epsilon)$. Setting variance $\epsilon$ sufficiently small produces weight matrices that are very close to identity. This makes forward pass analogous to multiple iterations of (noisy) QE applied to database descriptors [2]. Database QE has been shown to consistently improve retrieval quality by promoting local consistency [2, 13]. We observe a similar effect here, where even with near identity weights GCN improves base descriptors making GSS loss more effective.

Figure 1 shows the effect of training with our GCN architecture and GSS loss. Each row shows a query from the $\mathcal{R}$Oxford dataset [32] – a standard benchmark in image retrieval. GSS gradients are plotted against the similarity scores $s_{qi}$ between query $x_q$ and all database images $x_i \in \mathcal{X}$, relevant database images are colored in orange. Similarity threshold $\beta$ is set to 0.25 and is shown with a vertical dashed line. For each query we show progress at various stages of training from epoch 0 to 300, and compute average precision (AP) retrieval accuracy at each stage. From the figure it is seen that the initial similarity scores for query A in the top row are reasonably well separated, and most relevant examples are above the $\beta$ threshold. Our model is able to make quick progress here and further separate the relevant/not relevant examples into increasingly tighter clusters as learning progresses.

The bottom row of Figure 1 shows a less well separated query B. Here, our model is able to significantly improve the base descriptors, and still achieve near perfect separation gaining almost 20 points in mAP. We also clearly see that the GSS gradient increases linearly as scores move further away from $\beta$ in either direction. This is contrary to many existing objectives such as cross-entropy where gradients gradually go to 0 once predictions become close to target. We found this property to be instrumental to achieving the best performance. One possible explanation is that GSS puts a lot more emphasis on fully separating/collapsing the descriptors resulting in much tighter clusters.

### 3.3 Spatial Verification

The effectiveness of the GCN architecture depends on the quality of the adjacency matrix. Spatial verification [29, 2] is a commonly used approach to reduce false positives by applying robust verification with local feature matching. It is particularly effective in combination with global descriptors that are primarily designed for fast retrieval at the cost of false positives [2, 32]. While effective, computing local features is computationally expensive and can significantly slow down retrieval pipeline [29]. In this work we propose an efficient approach to incorporate spatial verification into our model.

The main idea is to apply verification in the offline training phase to refine the adjacency matrix $A_k$. During inference, however, verification is removed and initial retrieval for each query is done only with base descriptors. Interestingly, we found that once the GCN encodes information from the refined $A_k$ into database descriptors, the benefit of spatial verification can be effectively preserved *without* explicitly applying it to query during inference. This approach enables us to offload the computational burden to the offline phase, while still maintaining accuracy benefit during inference.

To refine the adjacency matrix, for each database image $x_i$ we first retrieve a set of candidate images $\mathcal{V}(x_i)$. Spatial verification is then applied to each image in $\mathcal{V}(x_i)$, and top-k images with highest verification scores are kept to get $\mathcal{V}_k(x_i)$. The refined set is used to compute the adjacency matrix:

$$a_{ij} = \begin{cases} x_i^\top x_j & \text{if } x_j \in \mathcal{V}_k(x_i) \vee x_i \in \mathcal{V}_k(x_j) \\ 0 & \text{otherwise} \end{cases} \tag{7}$$

where $\mathcal{V}_k(x_i)$ is the *verified* k-nearest neighbors of $x_i$. The rest of the GCN architecture is applied as before. The size of the candidate set $\mathcal{V}$ is a hyper parameter, and generally better results can be obtained with larger candidate sets particularly in cases where descriptor retrieval is not accurate. This, however, comes at the expense of additional computational cost.

### 3.4 Inference

Once the model is trained, given a new query $x_q$ we need to retrieve relevant images for $x_q$ from $\mathcal{X}$. In the offline phase, we make a forward pass through the GCN to compute updated database

Table 1: mAP retrieval results on INSTRE, $\mathcal{R}$Oxford and $\mathcal{R}$Paris (Medium and Hard) datasets. Spatial Verification section contains approaches that use spatial verification as part of pipeline.

| Method | mAP | | | | |
|---|---|---|---|---|---|
| | INSTRE | $\mathcal{R}$Oxford | | $\mathcal{R}$Paris | |
| | | Medium | Hard | Medium | Hard |
| GeM [33] | 69.1 | 64.7 | 38.5 | 77.2 | 56.3 |
| GeM+aQE [33] | 74.6 | 67.2 | 40.8 | 80.7 | 61.8 |
| GeM+DFS [19] | 81.1 | 69.8 | 40.5 | 88.9 | 78.5 |
| GeM+FSR [17] | 78.2 | 70.7 | 42.2 | 88.7 | 78.0 |
| GeM+DFS-FSR [18] | 77.9 | 70.5 | 40.3 | 88.7 | 78.1 |
| GeM+IME [45] | 82.3 | 70.4 | 45.6 | 85.0 | 68.7 |
| GeM+DSM [36] | - | 65.3 | 39.2 | 77.4 | 56.2 |
| GeM+DSM [36]+DFS | - | 75.0 | 46.2 | 89.3 | 79.3 |
| **GeM+GSS** | **89.2** | **77.8** | **57.5** | **92.4** | **83.5** |
| Spatial Verification | | | | | |
| GeM+aQE+DELF [28]-SV | 87.4 | 77.2 | 54.9 | 88.9 | 74.8 |
| GeM+DFS+HessAff-ASMK [40]-SV | - | 79.1 | 52.7 | 91.0 | 81.0 |
| HessAffNet-HardNet++ [27]+ HQE [20]-SV | - | 75.2 | 53.3 | 73.1 | 48.9 |
| **GeM+GSS$_\mathcal{V}$** | 90.5 | 79.1 | 62.2 | 93.4 | 85.3 |
| **GeM+GSS$_\mathcal{V}$-SV** | **92.4** | **80.6** | **64.7** | **93.4** | **85.3** |

descriptors $\tilde{x}_i$ for each image in $\mathcal{X}$. Then, to get an updated descriptor $\tilde{x}_q$ we incorporate query into the adjacency matrix, and make another forward pass through the GCN. However, multiplication with $A_k$ in each GCN layer introduces a dependency on $k$ instances in $\mathcal{X}$. This dependency grows at the rate of $k^L$ for a model with $L$ layers and can quickly make forward pass prohibitively expensive.

To deal with this problem we use an approximation where only first and second order neighbors of the query are retained. Formally, we define an approximate query adjacency matrix $A_k^q = (a_{ij}^q) \in \mathbb{R}^{(n+1)\times(n+1)}$:

$$a_{ij}^q = \begin{cases} x_q^\top x_j & \text{if } i = q, x_j \in \mathcal{N}_k(x_q) \\ x_i^\top x_j & \text{if } x_i \in \mathcal{N}_k(x_q), x_j \in \mathcal{N}_k(x_i) \\ 0 & \text{otherwise.} \end{cases} \tag{8}$$

Note that $A_k^q$ is highly sparse and repeated multiplication with this matrix only requires at most $k(k+1)$ descriptors from $\mathcal{X}$. Forward pass with $A_k^q$ can thus be done very efficiently by first caching the required descriptors, and then applying sparse matrix multiplications. Empirically, we find that approximate inference has negligible effect on accuracy vs running the full forward pass. This is consistent with previous findings for database-side QE [13], where large $k$ is used to construct the augmented database descriptors, but a much smaller $k$ is used during inference. The intuition is that the new database descriptors $\tilde{x}$ already encode extensive neighbor information making query augmentation less critical.

This approximate adjacency matrix is used to make a forward pass through the GCN as outlined in Section 3.1 to get the new query descriptor $\tilde{x}_q$. Retrieval is then done via inner product in the new descriptor space. It is important to note here repeat queries can be handled very efficiently and don't require forward passes through the GCN.

## 4 Experiments

**Datasets** We evaluate our model on five challenging public benchmarks. The popular Oxford [29] and Paris [30] datasets have recently been revised to include more difficult instances, correct annotation mistakes, and introduce new evaluation protocols [32]. The new datasets, referred to as $\mathcal{R}$Oxford and $\mathcal{R}$Paris, contain 4,993 and 6,322 database images respectively. There are 70 query images in each dataset, and depending on the complexity of the retrieval task evaluation is further partitioned into Easy, Medium and Hard tasks. In this work, we focus on the more challenging Medium and Hard tasks. We also evaluate our model on the INSTRE dataset [42] which is an instance-level image retrieval benchmark containing various objects such as buildings, toys and book covers in natural scenes. We follow data partitioning and evaluation protocol proposed by [19] with 1,250 query and 27,293 database images used for retrieval. Performance of all models is measured by the mean average precision (mAP).

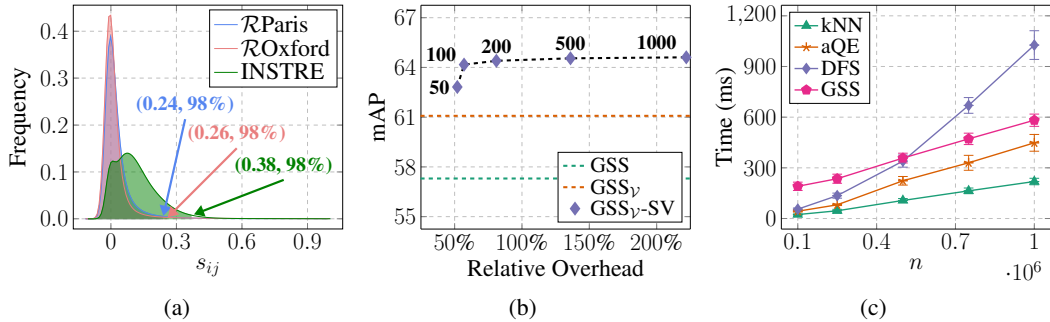

Figure 2: (a) $\beta$ selection using the 98'th percentile of the pairwise score distribution. (b) Relative inference run-time overhead vs mAP from applying SV to query on $\mathcal{R}$Oxford Hard. Number of verified candidates $|\mathcal{V}(x_q)|$ (see Section 3.3) is shown next to each point for $\text{GSS}_\mathcal{V}$-SV. We also show performance for GSS and $\text{GSS}_\mathcal{V}$ that don't apply SV to query. (c) Average per query inference run-time for GSS and several baselines. kNN corresponds to nearest neighbor retrieval with GeM. These experiments were done on the larger version of $\mathcal{R}$Oxford with 1M images.

**Baselines** We benchmark our GSS model against leading baselines including aQE [33], IME [45], and state-of-the-art similarity propagation methods DFS [19], FSR [17], DSM [36] as well as their combinations [18]. For spatial verification (SV) we compare against recent results including global-local hybrid model with DFS [40], DELF-based SV [28], and leading neural network model based on HessAffNet [27] with local feature bag-of-words pipeline [37]. To make comparison fair all models use the same GeM descriptors [33]. We use code and weights released by the original authors [1], and don't do any re-training or fine-tuning. Following authors' pipeline, multiple scale aggregation and discriminative whitening are applied to obtain 2,048-dimensional descriptor for each image. For spatial verification, we follow a standard pipeline of [7] to filter image pairs based on estimated inlier counts over aligned points of interest computed by RANSAC [11]. Deep local feature model (DELF) [28] is used for $\mathcal{R}$Oxford and $\mathcal{R}$Paris since it is trained specifically for landmark retrieval and used extensively in recent baselines. SIFT descriptors [25] are used for the more general INSTRE dataset.

**GSS** We implemented our approach using the TensorFlow library [1]. After parameter sweeps, two-layer $2048 \rightarrow 2048 \rightarrow 2048$ GCN architecture produced the best performance and is used in all experiments. To set the important $\beta$ parameter we note that only image pairs with sufficiently high similarity scores should be pushed closer together in the GSS loss. This leads to a general procedure where we first compute distribution of the pairwise scores $s_{ij}$, then set $\beta$ in the upper percentile of this distribution. We consistently found that using 98'th percentile worked well across all datasets. Figure 2a illustrates this procedure and shows pairwise score distributions for each of the three datasets together with selected $\beta$ values. Other hyper-parameters are set as follows: $\alpha = 1$, $k = 5$ for $\mathcal{R}$Oxford; $\alpha = 1$, $k = 5$ for $\mathcal{R}$Paris; $\alpha = 1$, $k = 10$ for INSTRE. All models are optimized using the ADAM optimizer [22] with default settings and weight initialization outlined in Section 3.2 with $\epsilon = 10^{-5}$. For spatial verification we use $\text{GSS}_\mathcal{V}$ to denote models that were trained with spatial verification (see Section 3.3), and $\text{GSS}_\mathcal{V}$-SV to indicate that spatial verification is also applied at query time. The size of the candidate set $|\mathcal{V}|$ is fixed to 250 for all datasets unless otherwise stated. All experiments are conducted on a 20-core Intel(R) Xeon(R) CPU E5-2630 v4 @2.20GHz machine with NVIDIA V100 GPU. Model training takes around 30 seconds for $\mathcal{R}$Oxford and $\mathcal{R}$Paris, and 10 minutes for INSTRE.

## 4.1 Results

Table 1 shows mAP retrieval results on all five datasets. From the table we see that GSS outperforms all baselines on each dataset. Notably the improvement is larger on the more challenging Hard tasks for $\mathcal{R}$Oxford and $\mathcal{R}$Paris, where our model achieves a relative gain of up to 11 mAP points or 24% over the best baseline. This demonstrates that the proposed pairwise loss combined with GCN neighbor encoding can learn a much better descriptor space, improving over the input GeM descriptors by up to 50%. Similar pattern can be observed from the spatial verification section of the table. Here, $\text{GSS}_\mathcal{V}$ also outperforms all baselines that use spatial verification. Furthermore,

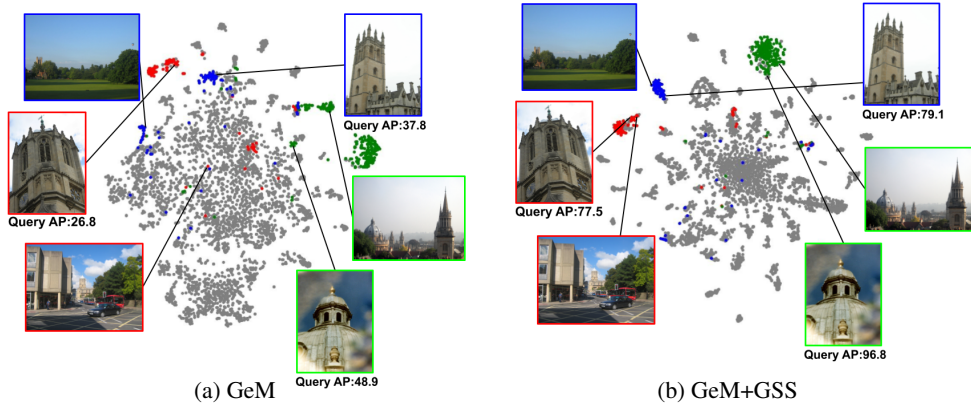

(a) GeM                    (b) GeM+GSS

Figure 3: Qualitative analysis on $\mathcal{R}$Oxford. GeM and GeM+GSS descriptors are plotted using PCA followed by t-SNE [26] projection to two dimensions. We show three example queries with corresponding relevant database images colored with red, green and blue. For each query, we display the query image (shown with AP score) and a hard relevant database image.

$GSS_{\mathcal{V}}$ consistently performs better than GSS gaining over 12% on $\mathcal{R}$Oxford Hard. These results indicate that our model can successfully encode information from spatially verified adjacency matrix into database descriptors, and improve performance *without* applying SV at query time. Further improvement can be obtained by also applying SV to query as shown by $GSS_{\mathcal{V}}$-SV but at additional cost. Figure 2b shows inference run-time overhead from applying SV to query as determined by the number of verified candidates. We see that additional gains in accuracy can be achieved by verifying up to 200 candidates retrieved by GeM. This, however, comes at a significant overhead of over 70% increase in run-time. $GSS_{\mathcal{V}}$ thus offers a principled way to leverage spatial verification without incurring additional inference cost.

Average query inference run-time in ms is shown in Figure 2c. To test the performance at scale we use the large version of the $\mathcal{R}$Oxford dataset with 1M database images. We vary database size from 100K to 1M, and record average query time over 100 restarts. During inference GSS requires kNN descriptor retrieval to compute the (approximate) adjacency matrix $A_k^q$, then forward pass through GCN to compute updated query descriptor $\tilde{x}_q$ and finally another kNN retrieval with $\tilde{x}_q$. The approximate inference procedure outlined in Section 3.4 is highly efficient, making the cost of the GCN forward pass independent of $n$. This is further evidenced by Figure 2c where GSS adds only a small constant overhead of around 140ms on top of aQE which also does two stages of kNN retrieval. In contrast, leading similarity propagation method DFS adds an overhead proportional to $n$ [19] which, as seen from Figure 2c, becomes significant as $n$ grows. While approximate and more efficient DFS inference has been proposed, it generally suffers from substantial decrease in accuracy [4].

Qualitative results on $\mathcal{R}$Oxford Hard are shown in Figure 3. Here, we use PCA and t-SNE [26] to compress the descriptors from GeM and our model to two dimensions and then plot them. We pick three queries and show their relevant database images in red, green and blue. For each query we also display the query image (shown with AP under each model) and one "hard" relevant database image. The hard images are visually dissimilar to the query and GeM is unable to retrieve them correctly. From the figure we see that our model learns much tighter clusters and significantly improves retrieval accuracy for each query. Specifically, it is able to place each of the shown hard database images into a tight cluster around the query even though there is little visual similarity between them.

## 5    Conclusion

We proposed a new unsupervised model for image retrieval based on graph neural network neighbor encoding and pairwise similarity separation loss. Our model effectively learns a new descriptor space that significantly improves retrieval accuracy while maintaining efficient inner product inference. Results on public benchmarks show highly competitive performance with up to 24% relative improvement over leading baselines. Future work involves further investigation into manifold learning as well as supervised version of our approach.

## Footnotes

[1] https://github.com/filipradenovic/cnnimageretrieval-pytorch

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
