[Reviews · NeurIPS 2019]

Reviewer 1



Strengths: State-of-the-art performance. The idea is novel and interesting, the evaluation is done properly, the results are significant for the field. The paper is clearly written. Experiments: Proper evaluation on three datasets (ROxford, RParis, INSTRE), comparison with state-of-the-art solutions. I strongly suggest acceptance.

Reviewer 2



Clarity: - the paper is globally well-written, but there are many unclear parts * parts mentioning the "manifold" without any prior explanation on what it is, e.g. lines 22~23, lines 27~28 * line 120: the said normalization procedure cannot ensure that the rows and columns will have unit norms (contrary to what is claimed). An trivial counter-example is the [[1,1],[1,0]] symmetric matrix. * line 147: not sure to understand * figure 2(b): the runtime overhead is relative to what? * how is implemented k-NN in all experiments? Novelty: - the method combines existing approaches together with a novel unsupervised loss. Namely, graph convolutional networks (GCN) that take as input the image descriptors and a fixed adjacency matrix defining pairwise similarities, while the novel GSS regression loss teaches the GCN how to cluster points based again on pariwise similarities on the output descriptors. - the paper claims to "introduce a fully unsupervised loss to optimize the model" [for image retrieval] (line 42), which is unheard of, and thus extremely impressive... but also deceiptful. As stated below, the success of the approach entirely depends on the initial image descriptors, which are state-of-the-art from [33] and trained with full supervision using a ranking loss. No experiments are given to investigate this crucial aspect. Quality: - experiments results are extremely good on multiple standard benchmarks, but some important points need to be clarified: - the proposed approach have 2 layers of GCN (eq. (3)), which is actually not deep at all (i.e. at best, second-order neighbors are used to augment the image descriptors). There need to be experiments to investigate this parameter. - similarly, there is no experimental proof that the GSS loss is actually driving the good performance. What happens when the GCN is applied with identity weights and null bias for 1 or 2 GCN layers? What would be the best is to plot the AP on multiple dataset as a function of the training iteration (0 --> 300). - isn't the approach actually just exploiting a bias due to the dataset construction? Specifically, we must recall that these datasets have been constructed by first gathering different clusters of related images, taking out few images from each cluster to form the queries, and then appending random distractor images (which are naturally unclustered). It is thus expected that a method explicitly clustering the dataset, like the proposed method, would achieve top performance precisely because it exploits this structure present in the dataset due to the construction procedure? But what would happen for real-life datasets? * a hint towards this hypothesis is that hyper-parameters need to be carefully tuned for each dataset separately (line 255). This is because the clusters in each dataset have different sizes and distributions. * one way to verify this hypothesis is to provide results for the datasets augmented with 1M distractor images. This would significantly hamper the clustering process (which is very easy and straightforward on small datasets) and would render things more realistic. * a related question is: what is the training time, as a function of the number of images in the dataset? - the convergence of this unsupervised approach, hence its good results, largely depends on the initial image descriptors. This paper uses descriptors from [33], but what would happen if these descriptors are corrupted by noise? Significance: - as far as i understand, the proposed approach is trained separately for each dataset, which strongly limits its general applicability - at test time, the paper need to embed the query into the adjacency matrix and make a forward pass through the GCN. As a matter of fact, this is more or less equivalent to some elaborate QE (and the same process has been applied to the whole database). Isn't this defeating the purpose of "directly encoding the neighbor information into image descriptors"? Because, in the end, the query needs to be processed even more than traditional QE (kNN search, then aggregation with neighbors using even more computations). This is very clear from Figure 2 (c).

Reviewer 3



Originality: - The approach is original. It uses GCN to learn and encode neighborhood information in image retrieval, that previously relied on mostly handcrafted QE/similarity propagation/spatial verification. This has the potential of learning higher-order relationships and more powerful representations. Quality: - The proposed technique appears sound and well researched. The proposed GSS loss is based on a simple intuition and is demonstrated to work well. The second-order approximation to the adjacency matrix during inference is reasonable. Clarity: - This paper is well written. The motivation and techniques are explained clearly, and implementation details are discussed. Ablation study on the most important parameter, $\beta$, is included. Significance: - The results show convincing improvements over state-of-the-art on INSTRE, ROxford and RParis. Although, it would be nice to see results on the augmented ROxford+1M and RParis+1M.

[Author Response · NeurIPS 2019]



Figure 1: (a) Retrieval performance vs number of GCN layers on $\mathcal{R}$Oxford Hard. (b) Training curves for a two-layer GCN model with GeM and R-MAC descriptors on $\mathcal{R}$Oxford Hard. (c) Binned distribution of pairwise similarity scores $s_{ij}$ for all three datasets.

We would like to thank the reviewers for taking the time to review our work and providing valuable feedback. Here, we
address main concerns brought up by each reviewer, and will incorporate minor feedback directly into the draft. We are
also in the final stages of refactoring our code repository, and will open source code for all experiments with the final
version of this draft.

**Reviewer 1** We thank the reviewer for the highly positive feedback and encouraging comments.

**Reviewer 2** To address the detailed questions raised by the reviewer, we ran additional experiments to further in-
vestigate the properties of our model. Due to time constraints most experiments were done on the $\mathcal{R}$Oxford Hard
dataset. Figure 1a shows the effect of adding more layers to the GCN with error bars from ten restarts with ran-
dom weight re-initialization. We were initially not able to optimize deeper GCNs and thus settled on two layers.
However, recently we discovered that adding residual connections (analogous to ResNet) between successive GCN
layers significantly improved optimization enabling to train much deeper models. From the figure it is seen that
adding layers slightly improves performance from 57.3 with two layers to around
57.6 with five layers. We suspect that further gains can be obtained with more
sophisticated optimization techniques and/or architectural modifications analogous
to residual connections that aid gradient back-propagation. Figure 1b shows retrieval
performance vs training epoch for a two-layer GCN architecture. We see that applying
two GCN layers without training (epoch 0) already significantly improves performance
of the base GeM descriptors from 38.5 to 51.2. Similar improvements were observed
for all other datasets, and we found that normalizing the adjacency matrix according to
Equation 2 (in the paper) was instrumental to obtaining this boost. Applying one GCN
layer with near identity weights is analogous to "weighted" database side QE, so our results indicate that appropriately
normalising the adjacency matrix is highly important for QE and should be further investigated. We also see that
training the model with the proposed GSS loss further improves performance by over six points. So both GCN and GSS
components are important and best results are generally obtained when the two are combined.

Table 1: mAP on $\mathcal{R}$Oxford 1M.

| Method | mAP |
|---|---|
| GeM | 22.7 |
| GeM+$\alpha$QE | 24.2 |
| GeM+DFS | 33.2 |
| GeM+FSR | 18.8 |
| GeM+DFS+FSR | 34.4 |
| **GeM+GSS (ours)** | **35.8** |

**Reviewer 3** We have been investigating how to set $\beta$ automatically, and believe that a promising direction is to use the
distribution of the pairwise similarity scores $s_{ij}$. Figure 1c shows score distributions for $\mathcal{R}$Oxford, $\mathcal{R}$Paris and INSTRE
datasets together with $\beta$ which was set to 0.25 for $\mathcal{R}$Oxford and $\mathcal{R}$Paris and to 0.45 for INSTRE. Here, we see that
good values for $\beta$ tend to be at the *tail* of the score distribution so only the most confident scores get pushed up. This
suggests a heuristic to automatically set $\beta$ by first computing the empirical cumulative distribution function (CDF) of
similarity scores, and then setting $\beta$ to the value where the CDF is sufficiently high such as 0.9. This works well for the
three datasets that we evaluated on, and we believe that it can be generalised to other datasets as well.

**Reviewers 2 and 3** Both reviewers mentioned varying base descriptors and larger 1M results. Figure 1b shows a
training curve for our model with R-MAC [12] image descriptors. R-MAC alone achieves 32.4 on $\mathcal{R}$Oxford which is
significantly lower than the 38.5 achieved by GeM. Applying GCN improves the accuracy to 43.6, and GSS optimization
produces additional five point gain pushing the accuracy to 49.3 which also outperforms all baselines. These results
suggest that our model can be effectively used with different base descriptors regardless of their performance. Table 1
shows results on $\mathcal{R}$Oxford 1M for our model and GEM-based baselines that report results on this dataset. Note that
these results are very preliminary as we only had several days to train the model on a much larger dataset. To fit the
optimization on the GPU we switched to batch training for the 1M data, where random samples of images were used to
compute the GSS loss gradients and update GCN weights. Training to convergence took approximately five hours vs
two minutes for the smaller version of $\mathcal{R}$Oxford. From the table we see that our model outperforms the best baseline
DFS+FSR by over one point. This indicates that our approach does generalise to the harder setting where the number of
distractors is significantly larger. However, as Reviewer 2 pointed out, finding meaningfull clusters is considerably
more difficult in this setting so we plan to focus on large scale applications in future work.

[Meta-Review · NeurIPS 2019]

The reviewers unanimously agree that this is an important paper that is clearly written and with significant contributions. Please make sure to address some of the reviewers' concerns that were not addressed in the rebuttal during the camera ready version. Namely R2's questions regarding dataset bias, training time, test time, and per-dataset tuning.